# Improved Quality of Life, Fitness, Mental Health and Cardiovascular Risk Factors with a Publicly Funded Bariatric Lifestyle Intervention for Adults with Severe Obesity: A Prospective Cohort Study

**DOI:** 10.3390/nu13114172

**Published:** 2021-11-21

**Authors:** John Francis Brazil, Irene Gibson, Denise Dunne, Lisa B. Hynes, Aisling Harris, Mustafa Bakir, Dylan Keegan, Brian McGuire, Mary Hynes, Chris Collins, Siobhan Foy, Suzanne Seery, Paul Bassett, Colin Davenport, Jennifer Jones, Francis M. Finucane

**Affiliations:** 1Bariatric Medicine Service, Centre for Diabetes, Endocrinology and Metabolism, HRB Clinical Research Facility, Galway University Hospital, H91 YR71 Galway, Ireland; MustafaF.bakir@hse.ie (M.B.); brian.mcguire@nuigalway.ie (B.M.); mary.hynes1@hse.ie (M.H.); Siobhanc.Foy@hse.ie (S.F.); Colin.Davenport@hse.ie (C.D.); 2Heart and Stroke Center, Croi, The West of Ireland Cardiac Foundation, Moyola Lane, Newcastle, H91 FF68 Galway, Ireland; irene.croi@hse.ie (I.G.); d21year84@hotmail.com (D.D.); lisah@croi.ie (L.B.H.); Aisling.croi@hse.ie (A.H.); dylankeegan20@gmail.com (D.K.); jennifer.jones@nuigalway.ie (J.J.); 3National Institute for Prevention and Cardiovascular Health (NIPC), Newcastle, H91 FF68 Galway, Ireland; suzanne.seery@gmail.com; 4School of Psychology, National University of Ireland, H91 TK33 Galway, Ireland; 5Department of Upper Gastrointestinal Surgery, Galway University Hospital, H91 YR71 Galway, Ireland; Chris.COllins@hse.ie; 6Statsconsultancy Ltd., Amersham HP7 9EN, UK; paul@statsconsultancy.co.uk

**Keywords:** severe obesity, bariatric, fitness, quality of life, anxiety, depression, diet, physical activity, structured lifestyle modification

## Abstract

Background: Lifestyle modification is the cornerstone of management for patients with severe and complicated obesity, but the effects of structured lifestyle programmes on quality of life, anxiety and depression scores and cardiovascular risk factors are not well-described. We sought to describe changes in self-reported quality of life and mental health-related outcomes as well as cardiovascular risk factors in patients completing a 10-week multidisciplinary lifestyle-modification programme. Methods: We conducted a prospective cohort study of all patients referred from our bariatric service who completed the programme between 2013 and 2019. In addition to weight, body mass index (BMI), blood pressure, HbA1c, lipid profile and functional capacity, we quantified health-related quality of life using the Dartmouth COOP Questionnaire and the European Quality of Life Questionnaire Visual Analogue Scale (EQVAS) and mental health using the Hospital Anxiety and Depression Scale (HADS). Results: Of 1122 patients who started the programme, 877 (78.2%) completed it and were included in per protocol analyses. Mean age was 47.3 ± 11.9 years, 66.9% were female, 34.8% were in full- or part-time employment and 69.4% were entitled to state-provided medical care. BMI decreased from 47.0 ± 7.8 to 46.2 ± 7.8 kg m^−2^ and weight decreased from 131.6 ± 25.5 to 129.5 ± 25.4 kg (both *p* < 0.001). There were significant reductions in anxiety and depression scores and improvements in all Dartmouth COOP domains. The EQVAS score increased from 52 ± 22 to 63 ± 19 (*p* < 0.001). Small but statistically significant reductions in LDL cholesterol, systolic blood pressure and HBA1c were also observed. Conclusions: Adults with severe and complicated obesity completing a specialised bariatric lifestyle-modification programme showed significant improvements in self-reported mental health and quality of life, in addition to reductions in cardiovascular risk factors.

## 1. Introduction

Consistent with international trends [1], the prevalence of severe and complicated obesity has increased in Ireland [2]. Most of the associated mortality from excess body weight is due to cardiovascular disease [3]. Similarly, the health economic consequences of obesity are driven primarily by increased cardiovascular disease [4,5]. Lifestyle modification is universally recognised as the cornerstone of the therapeutic approach to obesity and its complications, as recommended by several expert clinical groups [6,7,8]. Large, randomised trials have confirmed the benefits of structured lifestyle modification in treating [9] or preventing type 2 diabetes [10,11,12] and in the secondary prevention of cardiovascular disease [13]. However, there is limited higher level epidemiological evidence from trials and systematic reviews that such interventions improve health in bariatric patients. Indeed the efficacy of lifestyle interventions in this population falls short of the significant improvements observed in patients undergoing bariatric surgery, which has been shown to be effective in reducing mortality [14], morbidity [15] and healthcare costs [16] for affected individuals. Methodological heterogeneity in intervention design has limited the availability of meta-analyses in this cohort [17]. Consistent with previous findings [18], we have noted that although meal replacement programmes lead to substantial short-term weight loss and improvements in cardiovascular risk factors, attrition rates are very high [19] and weight loss is not sustained in the longer term [20], as noted by others [21]. There is increased recognition of the importance of combining dietary and physical activity components in structured lifestyle programmes [17,22] and of adopting a more holistic approach to outcome evaluation than just weight loss [23].

The negative impact of severe obesity on quality of life at all ages is well-established [24,25]. Depression [26] and anxiety [27] are common psychiatric comorbidities in affected individuals. There is good evidence that purposeful weight loss with lifestyle modification can reduce anxiety and depression scores [28,29], but whether structured lifestyle-modification programmes have this effect in patients with severe and complicated obesity has not yet been determined. Severe obesity is associated with higher levels of social deprivation [30], which could make recruitment to and formal evaluation of structured lifestyle programmes more challenging. In 2013, we deployed a structured lifestyle-modification programme for patients with severe and complicated obesity attending our regional bariatric service—“Changing Lifestyle with Activity and Nutrition (CLANN)”. This was modelled on the successful implementation in the UK of a nurse-led, family-based lifestyle intervention that was focussed on cardiovascular risk reduction in patients with (or at high risk of) cardiovascular disease [31], and which was replicated by members of our group for high-risk cardiovascular patients [32] and those with type 2 diabetes [33] in the west of Ireland. We have previously described changes in anthropometric, metabolic and cardiovascular risk factors in the cohort of bariatric patients recruited in the first two years of the study [34]. Here, we sought to describe changes in self-reported measures of quality of life, anxiety and depression and to provide updated information on anthropometric and metabolic outcomes in programme completers.

## 2. Materials and Methods

This was a single-centre prospective cohort study conducted in accordance with STROBE guidelines [35]. The study population included patients who were referred to our community-based structured lifestyle intervention between 2013 and 2019 and who were over 18 years old at the time of referral and had a body mass index (BMI) ≥ 40 kg m^−2^ (or ≥35 kg m^−2^ with an obesity-related co-morbidity). Patients for whom the intervention was deemed suitable were referred following assessment by the hospital-based multidisciplinary bariatric medicine team. Patients with cognitive impairment, uncontrolled hypertension (grade 3, >180/110 mmHg) [36], symptoms suggestive of ischemic heart disease or those who were unable to walk 10 m unassisted were excluded from the programme.

At the first programme visit, each patient underwent an individualised assessment by the specialist CLANN multidisciplinary team (physiotherapist, exercise specialist, cardiovascular nurse and dietician) to obtain baseline anthropometric data, medical history and relevant medication usage (statin, antihypertensive and antiplatelet drugs) and to identify motivation, barriers and facilitators of behavioural change. Weight was measured using a Seca^®^ 877 scale and height with a Seca^®^ Leicester stadiometer. Blood pressure was measured with an Omron^®^ 705IT oscillometric device. Self-reported dietary and physical activity patterns were recorded based on seven-day activity recall. An Incremental Shuttle Walk Test (ISWT) was used to quantify functional capacity by deriving an estimate of maximal metabolic equivalent of task (Est METmax) [37]. Self-reported anxiety and depression scores were quantified using the Hospital Anxiety and Depression Scale (HADS) [38]. Self-reported quality of life was measured using the European Quality of Life Questionnaire Visual Analogue Scale (EQVAS) [39] and the Dartmouth COOP Questionnaire [40]. All blood samples were analysed locally in the Galway University Hospitals’ Department of Clinical Biochemistry (certified to ISO 15189 2007 accreditation standard). HbA1c was measured with HPLC (Menarini^®^ HA8160 auto-analyser, Florence, Italy). Total cholesterol was measured using the CHOP-PAP method. High-density lipoprotein (HDL) cholesterol and triglycerides were measured using the enzymatic and the GPO-PAP methods, respectively (Roche COBAS^®^ 8000 modular analyser, Basel, Switzerland). Low-density lipoprotein (LDL) cholesterol was derived with the Friedewald equation [41].

At the first programme visit, individualised exercise prescription and risk stratification took place in order to ensure that each patient had adequate progression of exercise intensity over the duration of the programme. Thereafter, weekly group-based sessions lasting 2.5 h each took place over eight consecutive weeks. These consisted of an educational workshop combined with a physical activity class. Although physical activity and exercise sessions were delivered in a group setting, they were individualised to be realistic, reproducible and acceptable to each patient. Exercise was performed without any specialist equipment in order to encourage continuation of the activity beyond the duration of the programme. Specific attention was given to reducing sedentary behaviour. Healthy eating choices were informed by the European guidelines for cardiovascular prevention [42] and a target of weekly weight loss of 0.5 kg was encouraged through a cardioprotective diet with an energy deficit of 600 kcal/day. The educational component consisted of workshops specific to diet (healthy eating principles, portion control, food labelling), exercise, physical activity, cardiovascular health, stress management and psychological issues relevant to people with obesity.

The main emphasis of the programme was on lifestyle modification, with a strong focus on behavioural change, but with the ultimate aim of cardiovascular risk factor reduction rather than weight loss per se. Established motivational interviewing strategies were used throughout the programme in order to enhance self-efficacy in achieving goals [43]. Patients were also given a personal record card to use on a weekly basis to record their goals and track their progress in relation to weight, BMI, physical activity, blood pressure and lipid profile and, for patients with diabetes, HbA1c and glucose levels were also recorded. Blood pressure and lipid targets were based initially on the 2012 European Society of Cardiology (ESC) prevention guidelines [42] when the blood pressure target was 140/90 mmHg (140/85 mmHg in patients with type 2 diabetes) and the lipid targets were total cholesterol < 5 mmol/L, LDL cholesterol < 3 mmol/L and triglycerides < 1.7 mmol/L. When the guidelines were updated in 2016 [44], we sought an LDL cholesterol < 1.8 mmol/L in patients with established cardiovascular disease, and again in 2018 we revised our blood pressure target downwards in patients with diabetes to <130/80 mmHg [36]. At the last programme visit after 10 weeks, all of the measures that were obtained at baseline were repeated. The study was approved by the Galway Clinical Research Ethics Committee (the ethics committee for Galway University Hospitals). All patients provided written informed consent for their data to be used in these analyses.

The statistical analyses focused on changes in outcomes between the first programme visit and the follow-up measure after completion of the programme at 10 weeks. Changes in categorical variables were assessed by the McNemar test or, for rarer outcomes, the paired exact test. Changes in continuous variables where the changes in values were found to be approximately normally distributed were examined using the paired *t*-test. The Wilcoxon matched-pairs test was preferred for continuous variables where the changes in values between timepoints were not normally distributed. SPSS version 24 was used for all analyses.

## 3. Results

Of a total of 2835 patients seen in the Galway University Hospital Bariatric Service between 2012 and mid-2019, 1447 (51%) were referred to the Croi CLANN programme. Of these, 1127 (77.8%) attended the initial assessment and 877 of those participants (also 77.8%) completed the end-of-programme assessment and were included in the analysis. Baseline demographic characteristics of programme starters are shown in Table 1. Their mean age (± standard deviation) was 47.3 ± 11.9 years (range 16–77), 66.9% were female and 27.1% had not completed a secondary/high-school education. A total of 61% of patients were living with a partner. Ethnicity data were only recorded for 129 (11.5%) patients, 97.7% of whom identified as “White Irish”. A total of 41.2% of programme starters were in full- or part-time employment or were self-employed, and 69.4% of patients were entitled to a “General Medical Services” card, allowing them access to means-tested and state-sponsored medical care. There was high prevalence of type 2 diabetes (26.7%), hypertension (44.7%), depression (31.4%), sleep apnoea (19.1%), back pain (47.6%) and arthritis (35.6%). It was found that 5.3% of patients had had a previous cardiac event and 1.2% a previous stroke; 52.2% of programme starters had a “low” individual cardiovascular risk score according to the European Society for Cardiology (ESC) guidelines [45] while 10.5% were at “moderate”, 19.5% at “high” and 17.8% at “very high” cardiovascular risk.

The baseline and follow-up anthropometric and metabolic characteristics of programme completers are shown in Table 2. There were modest but statistically significant reductions in weight, BMI and waist circumference, as shown. A total of 12.6% of participants lost 5% or more of their body weight and 1.6% lost 10% or more; 31.2% of participants did not lose any weight. There were reductions in systolic and diastolic blood pressure. There were reductions in all components of the lipid profile (including HDL cholesterol) but not in the triglyceride:HDL cholesterol ratio, suggesting insulin sensitivity did not change. There was a 20.9% reduction in the number of participants who smoked, and a 25% increase in aerobic fitness according to the estimated METmax. There was a more than six-fold increase in the proportion of participants achieving their weekly target of 150 min of moderate-to-vigorous intensity exercise per week.

There were high levels of self-reported anxiety and depression at baseline, with 53.2% of participants scoring ≥ 8 on the HADS anxiety score and 47.4% scoring ≥ 8 on the depression score. There were reductions in both scores in programme completers, as shown in Table 2, such that the proportions of participants scoring ≥ 8 for anxiety and depression decreased to 42.5% and 26.2%, respectively, with relative reductions of 20.2% and 44.8%, respectively (all *p* < 0.001). The self-reported EQVAS score increased. There were reductions in the level of impairment reported by participants in each domain of the Dartmouth COOP Questionnaire, as shown.

HbA1c was available in 84.3% of participants with type 2 diabetes who completed the programme and this declined significantly from 57.0 ± 16.3 mmol/mol to 54.0 ± 15.0 mmol/mol, with a mean decrease of −3.0 mmol/mol (95% CI −4.1, −1.9; *p* < 0.001). The proportion of patients with diabetes with a HbA1c less than 53 mmol/mol increased by 20.6%, with similar changes in weight and BMI as the overall cohort. Changes in blood pressure were also similar, with a mean reduction in systolic blood pressure of 15.2 mmHg (17.6, 12.8), *p* < 0.001. There were no statistically significant changes in the lipid profile in participants with diabetes.

## 4. Discussion

In this prospective cohort analysis of patients with severe and complicated obesity who completed a 10-week lifestyle modification programme, in addition to improvements in anthropometric, metabolic and cardiovascular risk factors, we noted improvements in self-reported mental-health and quality-of-life measures. To our knowledge, this represents the largest single-centre cohort study of bariatric patients undergoing such an intervention for which these mental-health and quality-of-life outcomes have been assessed. The retention rate in the intervention was relatively high, with 77.8% of patients attending both initial and follow-up assessments. This may have been because patients “opt in” to the programme and participation is not compulsory, even though all patients attending the regional bariatric service are encouraged to attend. Other potential reasons for good programme retention and completion are flexible programme times and an emphasis on patient autonomy in individual goal-setting, as well as the encouragement of peer support among participants. These characteristics have been shown to enhance participant experience in other studies [46]. Notwithstanding this, 22.1% of the patients referred did not start the programme and the barriers to participation and completion need to be explored in future studies.

The reductions in body weight we observed were more modest than previously described in similar studies [47]; however, these interventions often had longer durations. The impact of weight loss-focused interventions on cardiovascular outcomes remains uncertain [48] and weight regain is common following these interventions [21]. However, weight loss per se is not the focus of our programme; rather, the intention is to promote participants’ awareness of cardiovascular and overall health and to empower them with a sound working knowledge of the key principles of a healthy lifestyle, including healthy eating and the attainment of adequate levels of physical activity to improve fitness and reduce cardiovascular risk. Future studies could quantify changes in body composition, such as increases in lean mass and reductions in fat mass, which are likely to have occurred during the programme but which would not necessarily lead to overall reductions in BMI [49]. The increases in estimated aerobic fitness are important and significant, given that low fitness is associated with increased cardiovascular mortality in patients with obesity [50] and that increased fitness can augment other health gains from weight loss in patients with severe obesity [51], as well as decreasing cardiovascular mortality, even with modest fitness gains [52].

Changes in blood pressure and lipid profiles occurred while maintaining baseline medication usage throughout the intervention; they thus did not occur as a result of confounding from intensification of antihypertensive or lipid-lowering therapy. Similarly, in the subgroup of patients with T2DM, those who completed the programme showed a significant reduction in HbA1c, with an increase in the proportion achieving what would be considered “good” glycaemic control, without any medication changes. Whether these benefits are sustained in the longer term remains to be seen, but our findings are consistent with the well-established role of lifestyle modification as the cornerstone of the therapeutic approach to severe obesity and diabetes [53]. Given that anxiety and depression are more prevalent in this patient group [54,55], we think that the observed reductions in anxiety and depression scores are relevant and important. They are consistent with evidence from systematic reviews that anxiety and depression scores decrease with structured lifestyle-modification programmes in patients with obesity [28,56], though it is noteworthy that in the Look-AHEAD trial, anxiety and depression scores deteriorated in both the intensive lifestyle and the control groups, and there was no difference in the prevalence of antidepressant medication usage or quality of life between the groups after 10 years of follow-up [57]. Even after substantial weight loss with bariatric surgery, early improvements in anxiety and depression scores [58] may not be sustained in the longer term [59].

Our study has some limitations, not least the absence of a control group, the relatively short duration of the follow-up and the inclusion only of patients who completed follow-up measures. Thus, we cannot make inferences about the efficacy and effectiveness of the intervention. Nonetheless our observations offer a basis with which to conduct more rigorous assessment of the intervention in a randomised controlled trial and they suggest the intervention is likely to be beneficial to most patients completing it. The participation and completion rates were relatively high and, while the findings may not be generalizable to all patients with severe and complicated obesity, very little information has been available up until now about the response to lifestyle intervention in Irish adults with severe obesity. Another limitation is that, although anthropometric measurements were carried out by trained health care professionals, they were not blinded to the status (pre- or post-) of participants and were also involved in intervention delivery, which may have introduced bias to waist circumference or fitness measurements, for example, however unintentionally. Only randomised trials with allocation concealment would overcome this limitation, which should be borne in mind in future studies. The study also has a number of strengths, not least the large size of the cohort, the consistency of the nature and duration of intervention delivery and the inclusion of important fitness, mental-health and quality-of-life outcomes. These are important observations for health care professionals delivering multidisciplinary care, for policy makers and ultimately for patients with severe and complicated obesity.

## 5. Conclusions

In a cohort of Irish adults with severe and complicated obesity attending a regional bariatric service, those who completed a 10-week, multidisciplinary, structured lifestyle-modification programme showed reductions in body weight and cardiovascular risk factors and improvements in aerobic fitness and diabetes control, as well as improvements in self-reported mental-health and quality-of-life measures. These findings warrant further study in randomised controlled trials with longer follow-up.

## Figures and Tables

**Table 1 nutrients-13-04172-t001:** Demographic characteristics of patients with severe obesity who started the CLANN structured lifestyle-modification programme.

Demographic Variable	*n* (Total 1122)	Proportion
Sex (*n* = 1122):		
Female	751	66.9%
Male	371	33.1%
Employment status (*n* = 1068):		
Employed full-time	300	28.1%
Employed part-time	71	6.7%
Self-employed	69	6.4%
Carer for family	197	18.8%
Student	49	4.6%
Unemployed	132	12.4%
Retired	118	11.0%
Permanently sick	22	2.1%
Temporarily sick	33	3.1%
Other reason not working	73	6.8%
Entitlement to GMS * (*n* = 1029):		
Yes	714	69.4%
No	315	30.6%
BMI ≥ 45 (kg/m^2^)	466	56.4%
40–45	217	26.2%
35–40	113	13.7%
30–35	28	3.4%
<30	3	0.4%
Current smoker	93	8.3%
Hypertensive	330	44.7%
Type 2 diabetes	299	26.7%
Dyslipidemia	411	53.5%
ESC risk categorisation:		
Low	571	52.2%
Moderate	115	10.5%
High	213	19.5%
Very high	195	17.8%
HADS—depression score (*n* = 725)		
≥8	344	47.4%
≥11	158	21.8%
HADS—anxiety score (*n* = 726)		
≥8	386	53.2%
≥11	223	30.8%

* GMS—General Medical Services, refers to the means-tested provision of state-sponsored medical care.

**Table 2 nutrients-13-04172-t002:** Baseline and follow-up anthropometric, metabolic, quality-of-life, anxiety and depression measures from 877 CLANN programme completers.

Variable	*n*	Baseline	Follow Up	Change [95% CI]		*p* Value
Weight (kg)	829	131.6 ± 25.5	129.5 ± 24.4	−2.0	(−2.3, −1.7)	<0.001
BMI (kg m^−2^)	827	47.0 ± 7.8	46.2 ± 7.8	−0.7	(−0.8, −0.6)	<0.001
Waist (cm)	795	137.8 ± 17.3	134.4 ± 17.1	−3.5	(−3.9, −3.0)	<0.001
SBP (mmHg)	815	130.7 ± 15.9	115.0 ± 17.7	−15.8	(−16.9, −14.6)	<0.001
DBP (mmHg)	814	85.2 ± 10.4	83.8 ± 10.2	−1.4	(−2.0, −0.7)	<0.001
Total cholesterol (mmol/L)	803	4.69 ± 1.03	4.54 ± 0.98	−0.15	(−0.20, −0.11)	<0.001
HDL cholesterol (mmol/L)	795	1.20 ± 0.31	1.17 ± 0.31	−0.02	(−0.03, −0.01)	<0.001
LDL cholesterol (mmol/L)	771	2.79 ± 0.93	2.69 ± 0.87	−0.10	(−0.14, −0.06)	<0.001
Triglycerides (mmol/L)	803	1.5 (.1, 2.0)	1.4 (1.0, 1.9)	0	(−0.1, 0.0)	0.002
Triglyceride:HDL ratio	794	1.25 (0.86, 1.85)	1.23 (0.82, 1.82)	−0.01	(−0.04, 0.00)	0.16
Estimated MET_max_	570	5.6 ± 2.1	7.0 ± 2.8	1.4	(1.3, 1.6)	<0.001
Current smoker	844	93 (11.0%)	74 (8.8%)	−2.3%	(−3.5%, −1.0%)	<0.001
Exercise target achieved	760	44 (5.8%)	281 (37.0%)	31.1%	(27.6%, 34.7%)	<0.001
HADS anxiety score	725	8.3 ± 4.5	6.8 ± 4.2	−1.5	(−1.7, −1.2)	<0.001
HADS depression score	726	7.3 ± 4.1	5.2 ± 3.9	−2.2	(−2.4, −1.9)	<0.001
EQ-VAS	633	52 ± 22	63 ± 19	11	(9, 12)	<0.001
Dartmouth COOP domains:						
Physical	699	3.6 ± 1.1	2.9 ± 1.1	−0.8	(−0.8, −0.7)	<0.001
Feelings	718	2.7 ± 1.2	2.4 ± 1.2	−0.4	(−0.4, −0.3)	<0.001
Daily activities	716	2.6 ± 1.1	2.2 ± 1.1	−0.4	(−0.5, −0.4)	<0.001
Social activities	715	2.4 ± 1.3	2.0 ± 1.2	−0.4	(−0.5, −0.3)	<0.001
Pain	717	3.1 ± 1.3	2.9 ± 1.3	−0.2	(−0.3, −0.1)	<0.001
Change	719	2.8 ± 0.7	2.2 ± 0.9	−0.6	(−0.7, −0.5)	<0.001
Overall health	720	3.5 ± 1.0	3.0 ± 1.0	−0.6	(−0.6, −0.5)	<0.001
Support	707	2.3 ± 1.4	2.1 ± 1.4	−0.2	(−0.3, −0.1)	<0.001
Quality of life	713	2.6 ± 0.8	2.3 ± 0.9	−0.3	(−0.3, −0.2)	<0.001

Data are presented as means ± standard deviation or medians [inter-quartile range] or number of participants (percentage). BMI: body mass index, SBP: systolic blood pressure, DBP: diastolic blood pressure, HDL: high-density lipoprotein, LDL: low-density lipoprotein, MET: metabolic equivalent of task, HADS: Hospital Anxiety and Depression Scale, EQVAS: European Quality of Life Questionnaire Visual Analogue Scale.

## Data Availability

Data are available from the corresponding author upon reasonable request.

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
