# Peer review of "Improved Quality of Life, Fitness, Mental Health and Cardiovascular Risk Factors with a Publicly Funded Bariatric Lifestyle Intervention for Adults with Severe Obesity: A Prospective Cohort Study"

_nutrients, 2021, doi:10.3390/nu13114172_

Round 1

Reviewer 1 Report

1.This is a single-centre prospective cohort study so the Materials and Methods should be changed into Subjects and Methods. 

2.The SPSS 24 should be given the serial number and corporation name.What is the significance level in the statistical analyses?

3.The table 2 should be modified into three line table and and McNemar test, or the paired exact  test value or Wilcoxon matched-pairs test value.

Author Response

Nutrients manuscript ID 1451986:  Author Responses to Reviewers’ Comments.

Reviewer 1.

Comment 1:

“This is a single-centre prospective cohort study so the Materials and Methods should be changed into Subjects and Methods.”

Response 1:

We thank the reviewer for their careful and constructive consideration of our manuscript, which has allowed us to make some improvements.  We have changed the “materials and methods” section name to “subjects and methods”, as requested.

Comment 2:

“The SPSS 24 should be given the serial number and corporation name. What is the significance level in the statistical analyses?”

Response 2:

In fact, our statistician used Stata (version 15.1) for all analyses – we have included “Stata Statistical Software (Release 15.1, StataCorp LLC College Station, TX) was used for all analyses.” in the methods and appreciate the opportunity to provide this correction.

We have stated in the revised manuscript that “a p value <0.05 was considered statistically significant”.  However, we did not perform a correction for multiple comparisons.  In fact, all of the statistically significant differences we describe in the paper have a p value ≤0.002.

Comment 3:

“The table 2 should be modified into three line table and a McNemar test, or the paired exact test value or Wilcoxon matched-pairs test value.”

Response 3:

We have prioritised the presentation of clinical data in Table 2, in order to convey to the reader the changes in clinical variables through the program.  We feel that this is the most accurate and fair way to represent the statistical analyses.  In our previous description of the statistical methods, we had stated that “Changes in categorical variables were assessed by the McNemar test, or the paired exact test, for rarer outcomes.  Changes in continuous variables, where the changes in values were found to be approximately normally distributed were examined using the paired t-test.  The Wilcoxon matched-pairs test was preferred for continuous variables where the changes in values between timepoints were not normally distributed.”  We are confident that this provides adequate transparency and clarity around our statistical methodological approach, but we are aware that we may be missing the point that the reviewer is making.  We do not understand how we could modify table 2 into a three-line table, but would be happy to consider amending it if this is necessary and we understand what amendments are required, and why.

Reviewer 2 Report

In this single center study, Dr. Brazil and colleagues investigated the impact of a predefined lifestyle intervention programme on the mental health and quality of life of patients with obesity. They concluded that there were significant improvements in self-reported mental health and quality of life in patients with obesity following the 10-weeks implementation of their programme. Overall, the manuscript is nicely written and could be of interest. However, several issues occur and need to be addressed by the authors:

  • The title and the abstract seem to be misleading. The authors indicated that they also assessed CV health and CV risk in this manuscript. However, I could not find anything about them except for some metabolic parameters and smoking status. If those are indeed the CV risk meant by the authors, they need to mention "metabolic parameters" clearly in the title, instead of using CV health. Otherwise, the authors need to assess the CV health, for example by performing ECG and echocardiography, although I doubt there will be visible change in 10 weeks. 
  • The abstract said "improvements in self-reported mental health and quality of life in addition to reductions in cardiovascular risk." However, the result section of the abstract did not say anything about it. Please add some information supporting this claim or remove this statement about CV risk reduction.
  • I do feel that the changes observed in 10 weeks of the study are clinically meaningless. A reduction of 1-2 kg per 2.5 months are very slow and I am not sure if it would indeed lower the long-term CV morbidity and mortality. Moreover, as shown in Table 2, the reduction of the cholesterol levels are also small compared to pharmacological interventions. I suspect that this statistical significance was because of the large sample size only. Please comment on this and I am curious if there is any way to justify the long-term benefit of such a small reduction. 
  • Previous studies have shown that lifestyle modifications only modestly reduced weight and only 5.3% maintained weight loss (https://www.ahajournals.org/doi/full/10.1161/CIRCRESAHA.116.307591). The authors need to convince the readers why do they think that lifestyle modifications are still relevant in the long term as compared to surgery? Especially, by looking at the small reductions in this manuscript, the authors need to provide a strong background to justify the potential significance of this study. 
  • Related to my previous comment, it would be important to compare bariatric surgery and lifestyle modification in terms of weight reduction, disease prevention and CV health. Bariatric surgery has been shown to reduce MACE in a meta-analysis (https://doi.org/10.3390/nu13103568) and I think the authors need to discuss this aspect in the introduction so that people believe that lifestyle modifications are relevant for morbid obesity as studied in the current manuscript. 
  • Lines 72-74: "Here, we sought to describe changes in self-reported measures of quality of life, anxiety and depression and to provide updated information on anthropometric and metabolic outcomes in programme completers." Why was CV risk/health not shown anymore here? Please be consistent whether the CV aspect is something that the authors aim to cover in this manuscript. Please adapt the whole manuscript as well if needed. 
  • Please add a flowchart of the study design, including all the included and excluded participants, with the selection criteria used.
  • Line 159: Please specify the grade of the hypertension.
  • Line 161: Please specify the "cardiac event" meant there.
  • Table 1 needs to be expanded to include baseline data as described in the text, such as comorbidities, CV risk score, HADS, EQVAS, etc.
  • I am curious how to make sure that the EQVAS change before and after the intervention was not due to psychological subjectivity, for example because the patients "feel" that they are better than before (suggestive / placebo effect)?
  • I think the authors need to explain briefly about the reason of adding information about education, marital status and employment. I don't see any discussion about those demographic backgrounds. I am not sure if they are relevant unless the authors used those data somewhere in this manuscript. 
  • It is necessary to add the CV risk score before and after treatments (e.g., in Table 2) if the authors still want to include CV risk in this study. 

Author Response

Please find response attached.

Round 2

Reviewer 2 Report

Thank you for the responses to my previous comments. The authors’ answers are well taken and I have no further comments.

Regarding the unused demographic information, I personally do think that they are not needed and can be removed. Adding unused information could clutter the manuscript.

Author Response

We are grateful for the considered analysis of the reviewer and their significant input into the overall improvement of the manuscript.

On foot of these recommendations we have removed the demographic data relating to employment status and relationship status from table 1. We have retained information relating to employment status and entitlement to public healthcare as we believe that this research may be useful to inform the provision of healthcare services to people living with obesity in Ireland.